# Research on Wear Resistance of AISI 9310 Steel with Micro-Laser Shock Peening

**Xianhao Li [1], Liucheng Zhou [1,2,\*], Tianxiao Zhao [1], Xinlei Pan [1] and Ping Liu [3]**

1    Institute of Aeronautics Engine, School of Mechanical Engineering, Xi'an Jiaotong University, Xi'an 710038, China
2    Science and Technology on Plasma Dynamics Laboratory, Air Force Engineering University, Xi'an 710038, China
3    Institute of Aviation, Chongqing Jiaotong University, Chongqing 400074, China
\*    Correspondence: happyyzlch@163.com; Tel.: +86-133-2454-3606

**Abstract:** Improving the wear resistance of turbine engine drive components is crucial. This study presented a new Laser Shock Peening (LSP) technique: Micro-Laser Shock Peening (Micro-LSP) technology for surface modification and strengthening of AISI 9310 steel. The effects of different pulse energies (50 mJ, 150 mJ, 200 mJ) on surface morphology, mechanical properties, and wear behavior were investigated. The results showed that the Micro-LSP treatment reduced the wear rate by 56% to 74%. The dimpled structure induced during the strengthening process increased the surface roughness and reduced the contact area; moreover, the coefficient of friction (COF) was reduced. The treatment also had the effect of reducing the wear rate by collecting abrasive debris and changing some of the sliding wear into rolling wear. The reduced wear rate was a result of the combined effect of the dimpled structure and the hardened layer. In addition, a deeper hardened layer also slows down the onset of wear behavior. Micro-LSP technology offers completely new methods and possibilities for wear reduction.

**Keywords:** Micro-Laser Shock Peening (Micro-LSP); AISI 9310 steel; frictional wear; wear resistance





## 1. Introduction

AISI 9310 steel has the advantages of low costs and high performance and has been widely used in mechanical engineering and the aerospace industry. It is one of the most widely used gear materials for helicopter gearboxes and turbine engine transmission components [1]. Finding ways to significantly improve its wear resistance has been the focus of researchers. The use of many traditional techniques can improve this wear resistance, such as the use of ion implantation, chromium plating [2–5], and the addition of coatings [6–10].

However, these techniques also have significant disadvantages. For example, when using coating techniques, the joints could crack and flake due to poor bonding. This leads to a reduction in material performance. Laser Shock Peening (LSP) is an effective surface treatment technology that offers significant advantages over other techniques. LSP generates a large amount of high-pressure, high-energy plasma during the strengthening process, which can introduce residual compressive stresses and hardened layers onto the surface of the specimen. It can increase the surface microhardness and refine the grain of the metal material to improve fatigue [11–13], wear resistance [14–16], tensile resistance [17–19], and corrosion cracking resistance [20–22]. Due to its excellent properties, many scholars devoted themselves to LSP-related research in recent years. For example, Shi [23] et al. used the LSP technique for rail-welded joints. The results showed a reduction in wear rates of 5.1% and 10.1% in the weld area and heat-affected zone after strengthening, respectively. Both the average crack density and the average crack length were smaller than in the untreated area. Tong [24] et al. carried out LSP treatment of TC11 at high temperatures, and studied the wear performance at different temperatures and loads. It

was noted that the wear resistance of LSP-reinforced specimens at 500 °C was reduced by 29.99% compared to the base material. The main reason for this was the reduction in grain size. The frictional properties of the magnesium alloy ZK60 were investigated by Guo [25] et al. He used the LSP technique to treat the material, and results showed that abrasive wear was the main wear mechanism after LSP treatment. The wear rate was reduced by 17.6% compared to the base material. In summary, the methods used by most researchers were extremely limited in terms of the wear rate of the specimens. In addition, LSP technology also had some disadvantages. For example, the slow frequency made machining times too long when machining large areas. Furthermore, the large spot diameter of LSP made it impossible to machine complex parts in detail in their corners. This study proposes a new LSP technology that is called Micro-Laser Shock Peening (Micro-LSP). This processing method also reinforced specimens via laser-induced shockwave. However, the difference with LSP technology is that the Micro-LSP technology uses a lower pulse energy (less than 300 mJ) and smaller pulse duration (less than 10 ns) [26,27]. Moreover, an absorbent layer is not required during work. This feature offers a significant reduction in processing cost. In addition, Micro-LSP technology has better performance for wear and tear resistance. This study showed that the wear rate can be reduced by approximately 74% after Micro-LSP treatment.

This study aimed to investigate the effect of Micro-LSP treatment on the wear resistance of AISI 9310 steel. The surface and abrasion marks with different pulse energies were analyzed with an optical microscope (OM), a scanning electron microscope (SEM), and an energy dispersive spectrometer (EDS). Surface profile, shape, roughness, and wear rate were measured. The wear mechanism of the increased wear resistance of AISI 9310 steel after impact with Micro-LSP was fully revealed.

## 2. Materials and Experiments

### 2.1. Experimental Material and Processing Parameters

The sample size was 10 mm × 10 mm × 5 mm, and was obtained from a commercial AISI 9310 steel board with a wire-cutting method (Ming Hang, Xi'an, China). Both sides of the AISI 9310 steel had been carburized. The carburizing process involves placing the raw material in a furnace at 900 °C for 2 h; then, it is transferred to a cooling chamber and cooled with argon gas for 1h, kept at 800 °C for 2 h, and then diverted to a 200 °C oil bath for cooling before finally being held at 150 °C for 2 h in air. The carburizing layer depth was approximately 1.5 mm. Polishing of the sample used silicon carbide paper with a grain number from 800 to 2000 before LSP treatment. Final cleaning was carried out in deionized water. The metallographic organization is shown in Figure 1a. The microstructure was mainly composed of acicular martensite and cytosolic austenite, and a small amount of carbide. The surface XRD analysis is shown in Figure 1b. The chemical composition of AISI 9310 steel was listed in Table 1.

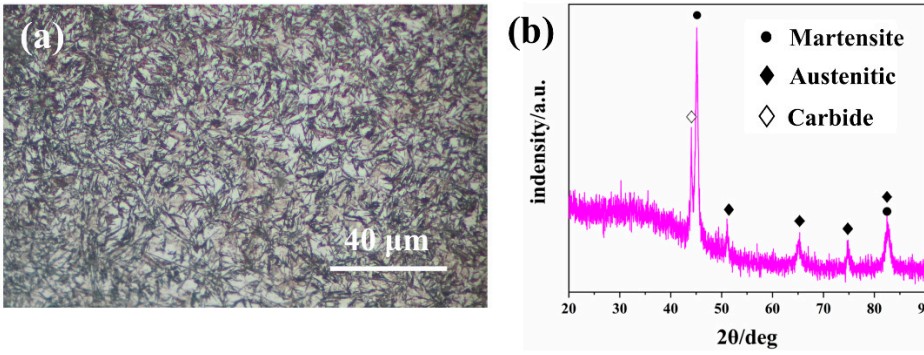

**Figure 1.** Schematic of (**a**) the metallographic organization and (**b**) the surface XRD analysis of AISI 9310 steel.

**Table 1.** Chemical composition of AISI 9310 steel (wt.%).

| Elements | Ni | Cr | Mn | Si | Mo | Cu | C | S | Fe |
|---|---|---|---|---|---|---|---|---|---|
| Contents | 3.24 | 1.23 | 0.63 | 0.26 | 0.12 | 0.12 | 0.11 | 0.005 | Bal. |

### 2.2. Micro-Laser Shock Peening Process

Figure 2a shows the process of Micro-LSP. It was carried out with an Nd:YAG laser device (Tyrida, Xi'an, China). The specific parameters of the strengthening process are shown in Table 2. Figure 2c illustrates the "Z" pattern machining method and processing direction. Strengthened specimens are shown in Figure 2b. During intensification, the laser beam was always perpendicular to the sample surface, and the specimen was restrained with a water-protective layer. No protective layer was used in the process. For the studied effect of different impact energy intensities on the wear resistance of AISI 9310 steel, three different pulse energies of 50 mJ, 150 mJ, and 200 mJ were set.

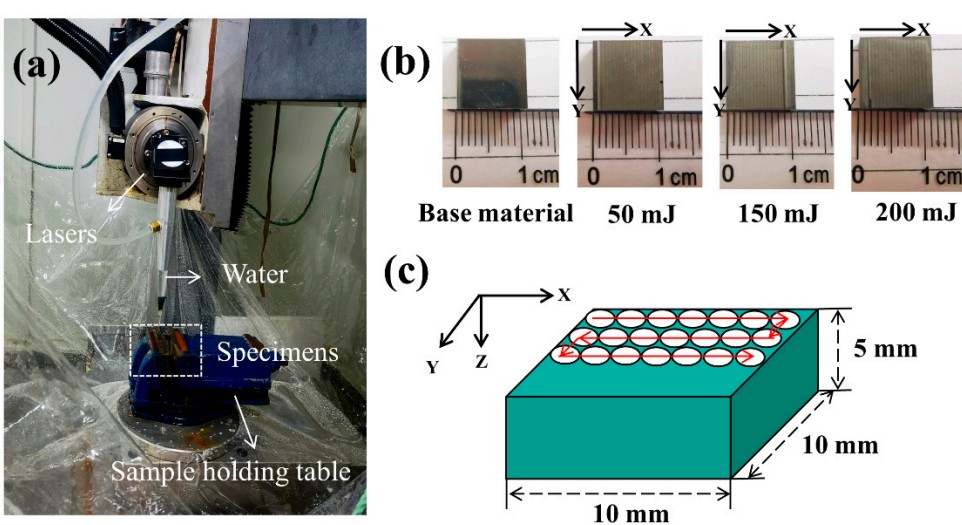

**Figure 2.** Schematic of (**a**) Micro-LSP process, (**b**) sample after strengthening, (**c**) "Z" pattern.

**Table 2.** Detailed parameters of the Micro-LSP process.

| Wavelength | Pulse Width | Spot Diameter | Overlapping Rate | Repetition |
|---|---|---|---|---|
| 532 nm | 8 ns | 0.5 mm | 50% | 500 Hz |

### 2.3. Frictional Wear Test

Reciprocal wear tests were carried out at room temperature using UMT-3 friction (CETR, Campbell, CA, USA) with the following parameters: $Si_3N_4$ with a diameter of 8 mm for the friction sub-material, a displacement amplitude of 8 mm, a normal load of 20 N; the friction time was 20,000 s, the velocity was 8 mm/s. Meanwhile, Figure 3 presented the processing of the wear process. All wear tests were performed in the air, and no lubricant was added. Three identical wear tests were carried out on each specimen. The wear volume was measured from an optical microscope (Keyence, Osaka, Japan) and used to calculate the wear rate. The 3D morphology, profile data, and surface roughness of the wear marks were measured using software (SuperView 2021 (64-bit) 1.0, SuperView W1, Shenzhen, China).

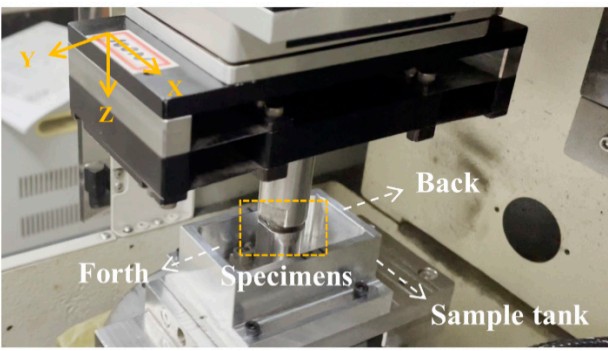

**Figure 3.** Schematic of friction wear process.

*2.4. Measurement of Microhardness and Surface Roughness*

The microhardness on the surface and in depth were measured using a Vicker′s indentation test machine (Taiming, Shanxi, China). The indentation load was 1000 g, and the holding time was 15 s. For each point, the average microhardness value was determined from the measured data of the four indentations.

*2.5. Microstructure Observation*

The surface appearance of AISI 9310 steel was observed with a scanning electron microscope (Zeiss, Wetzlar, Germany) and optical microscope. Furthermore, an energy dispersion spectrum (Zeiss, Wetzlar, Germany) was carried out to analyze the chemical composition of Micro-LSP surfaces and abrasion marks. Before using the energy dispersion spectrum for analysis, the samples were sanded with 2000-grit sandpaper and polished on a polishing machine (Veiyye, Laizhou, China). The microstructure was etched on the surface for 10 s–15 s using a 4% alcoholic nitric acid solution, and observed in a metallographic microscope (Zeiss, Wetzlar, Germany). The surface phase structure of the specimen was examined using X-Ray diffraction (Rigaku, Tokyo, Japan). The equipment used Cu targets as a source of X-ray generation. Tube current was 40 mA, the tube voltage was 40 kV, and the scanning angle range was 20° to 90°. The step size used was 0.02° with 0.245 s per step.

**3. Results**

*3.1. Surface Morphology and Roughness*

Figure 4a–d show the morphology of the surface treated with different pulse energies. The axes in Figure 4a illustrate the laser path during machining. The surface of the untreated specimen was smooth and had a metallic luster. After Micro-LSP treatment, the surface became darker and rougher, and lost its metallic luster. Periodically arranged dimpled structures on the surface could also be observed. The surface appeared the same after the different pulse energy treatments, and the surfaces were all slightly reconstructed. Figure 4e–h show the 3D morphology after Micro-LSP treatments. The axes in Figure 4e illustrate the laser path used during machining. It was similar to that observed in Figure 4a–d, where regularly arranged dimples could be observed on the surface. It was the mark left by the laser after impact on the surface of AISI 9310 steel. The shape and size of the light spot can also be observed in Figure 4b. Circular spots with a diameter of 0.5 mm were repeated at a 50% lap rate, which was consistent with the spot parameters. It was also observed that the roughness of the Micro-LSP treated specimens increased significantly compared to the untreated specimens. To quantify the effect of different pulse energies on the surface properties, Figure 4i shows the cross-sectional profile along the white dashed line, and Figure 4j shows the surface roughness (SA). Figure 4i shows that the depth of dimples formed on the surface of Micro-LSP-treated specimens with different energies were about 3.1 μm, 3.2 μm, and 3.5 μm. The greater the energy, the greater the depth of the dimples. It was also confirmed by the statistical surface roughness in Figure 4j.

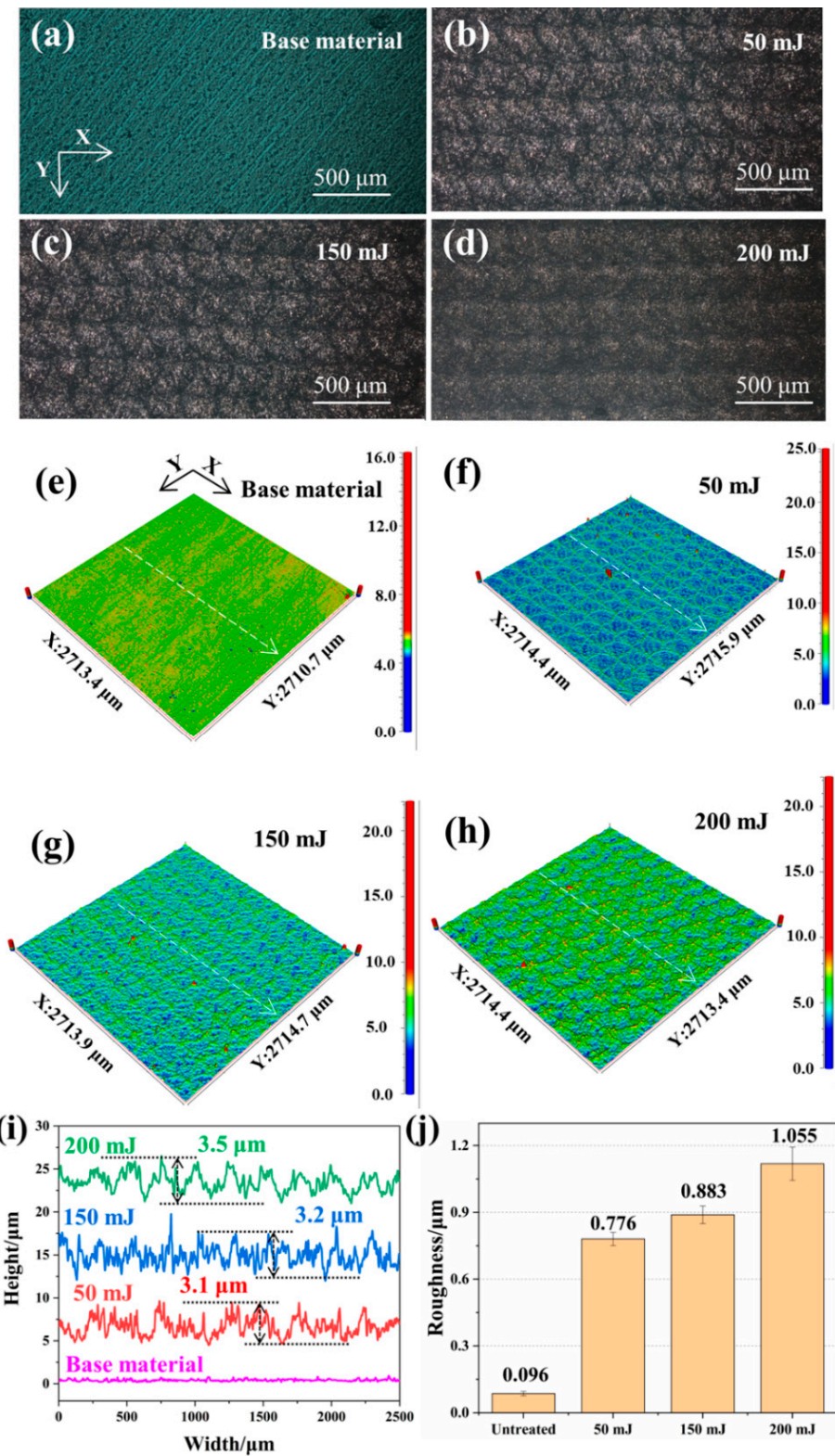

**Figure 4.** OM morphologies of (**a**) base material; (**b**) 50 mJ; (**c**) 150 mJ; (**d**) 200 mJ. 3D morphologies of (**e**) base material; (**f**) 50 mJ; (**g**) 150 mJ; (**h**) 200 mJ. The (**i**) cross-sectional profiles and (**j**) surface roughness of various Micro-LSP treatments.

### 3.2. Surface Profile

To further analyze the morphology and condition of the surface of AISI 9310 steel, SEM and EDS were carried out on the surface. Figure 5 shows the surface morphology after

different energy treatments using SEM. The axes in Figure 5a illustrate the laser path during machining. The results show that the laser processing energy had a significant effect on the surface topography. As the laser energy increased, the surface undulations became more pronounced. At small energies of 50 mJ, the traces left by the light spots can be observed. Then, as the laser energy increased, the surface of the specimen was completely ablated and lost its metallic luster. No visible traces of light spots were observed in the 200 mJ treated specimens of Figure 5g. In addition, laser deposition products and traces of remelting were found in Figure 5b,e,h. They were irregularly distributed on the surface, and covered the entire reinforced surface. It was similar to the surface of the reinforced LSP. Traces occurred in large numbers on the surface, and formed a continuous and dense multilayered organization. In addition, some studies had shown defects (cracks and holes) forming on the surface after LSP treatment [28,29]. These were closely related to the power density and the microstructure of the sample. Excessive pulse energy may cause the defect to change from a crack to a hole. However, no widespread defects were found in the strengthened specimens in Figure 5. This may be related to the low pulse energy, which did not reach the damage threshold.

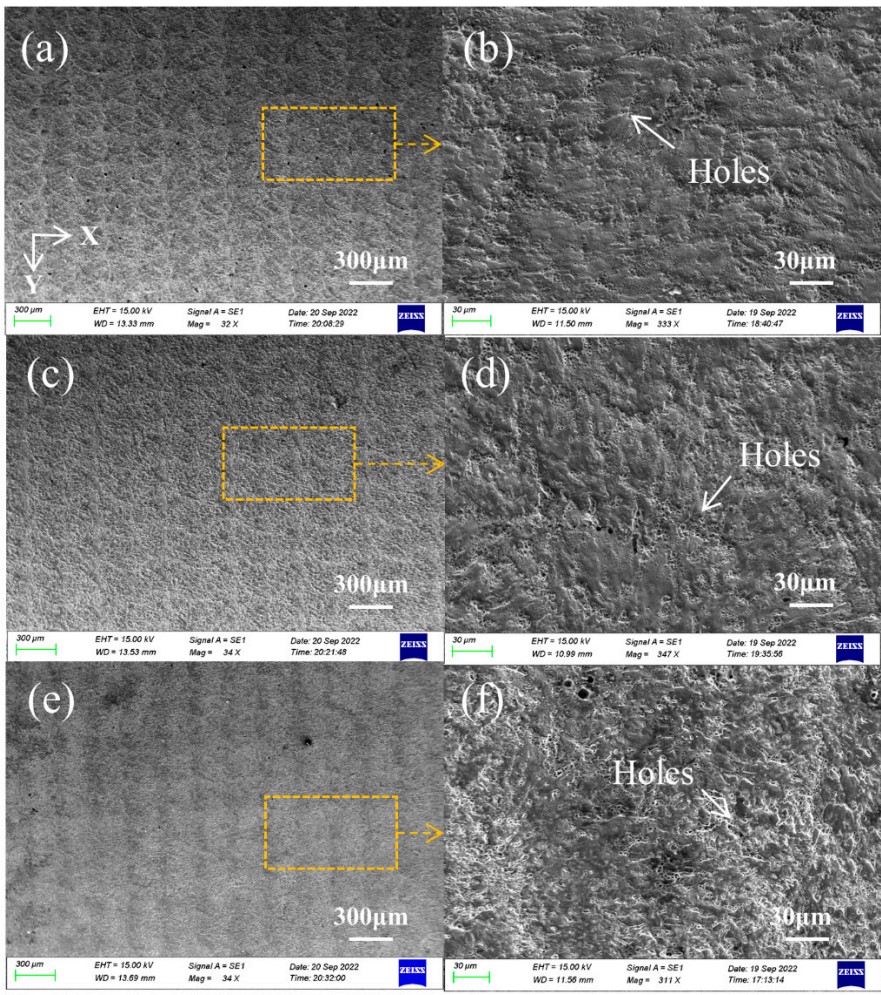

**Figure 5.** SEM morphologies after different energy treatments. (**a**,**b**) 50 mJ; (**c**,**d**) 150 mJ; (**e**,**f**) 200 mJ.

Figure 6 shows the distribution of the elemental content of oxygen on the surface after different energy treatments. The specific values of the elemental content are given in Table 3. The surface oxygen content of AISI 9310 steel was 1.33%, 15.96%, 19.77%, and 24.99% for the matrix for 50 mJ, 150 mJ, and 200 mJ treated steel, respectively. The surface oxygen content of the Micro-LSP-treated specimens was significantly higher compared to that of the matrix. This proves that the surfaces of the Micro-LSP treatment were the

same as the LSP treatment: a remelting oxide layer was formed [30–32]. The formation of the oxide layer was probably due to the reaction of the material surface with the oxygen provided in the water curtain during the Micro-LSP treatment. The dense multilayer rough structure observed in Figure 5b,d,f was mainly composed of oxide layers. In addition, EDS analysis showed that the concentration of elemental oxygen on the surface increased with pulse energy. This indicates that at higher energies, more elemental oxygen was involved in the reaction on the surface of the specimen, and was left on the surface of the specimen after remelting and cooling.

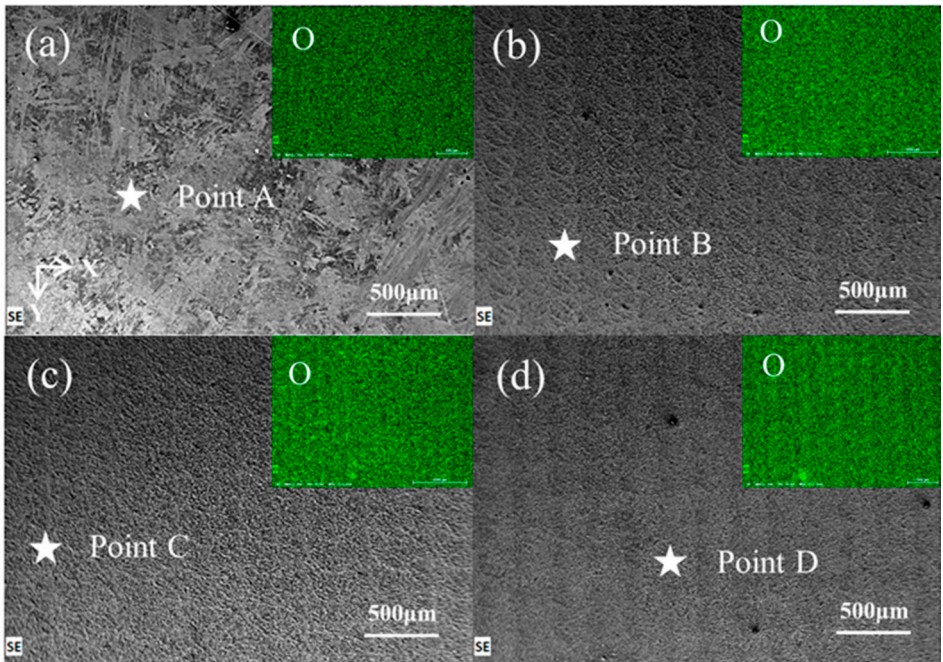

**Figure 6.** EDS morphologies after different energy treatments. (**a**) base material; (**b**) 50 mJ; (**c**) 150 mJ; (**d**) 200 mJ.

**Table 3.** Elemental content (wt.%).

| Element | O | Fe | C | Ni | Cr | Mn |
|---|---|---|---|---|---|---|
| Untreated | 1.33 | 88.04 | 5.88 | 2.97 | 1.27 | 0.34 |
| 50 mJ | 15.96 | 72.68 | 5.49 | 2.75 | 1.48 | 0.58 |
| 150 mJ | 19.77 | 69.59 | 5.76 | 2.88 | 1.33 | 0.49 |
| 200 mJ | 24.99 | 63.55 | 5.21 | 2.96 | 1.57 | 0.53 |

*3.3. Microhardness Analysis*

Hardness reflects the ability of the local volume of the material to resist deformation, and also influences the wear resistance of the material. The microhardness of the cross-section can, therefore, be used to assess the gradient of plastic deformation and the depth of the hardened layer after Micro-LSP treatment. Figure 7 shows the surface hardness and cross-sectional hardness of AISI 9310 steel after different pulse energy treatments. The surface hardness increased by 26%, 39.9%, and 48.3% after the 50 mJ, 150 mJ, and 200 mJ treatments, respectively, compared to the substrate. In addition, the depth of the hardened layer increased with increasing pulse energy. The depths at different pulse energies were approximately 400 μm, 500 μm, and 600 μm. Hardness results showed that the higher the pulse energy was, the greater the increase in hardness. The depth of the hardened layer was also greater. It could be easily understood that more laser energy means higher pressure. At the same time, this phenomenon was mainly attributed to the process-hardening effect introduced during the LSP process. As the depth gradually increased, the force of the shock wave induced by the laser diminished. This was the main reason for the drop in

hardness from the surface to around 800 MPa at 600 μm. In other words, the thickness of the hardened layer caused by a laser with a pulse energy of 200 mJ is approximately 600 μm. This hardened layer depth was determined with the relative Hugoniot elastic limit (HEL) given by Johnson and Rohde [33].

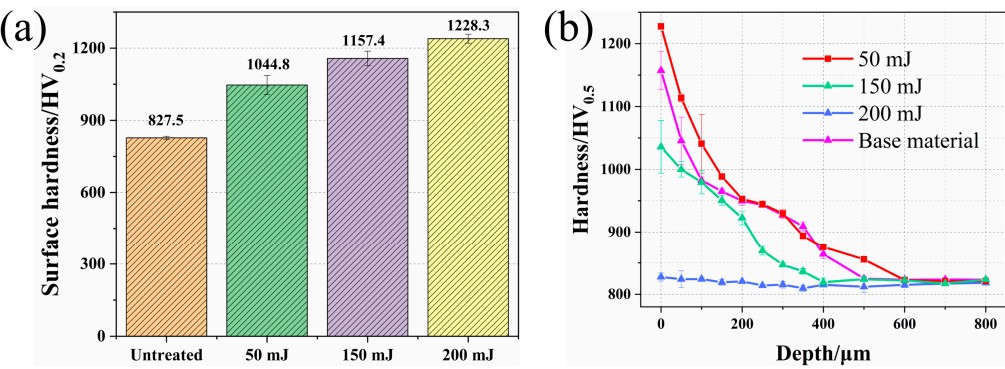

**Figure 7.** Hardness distribution after different energy treatments. (**a**) surface; (**b**) cross section.

### 3.4. XRD Analysis

Figure 8 shows the XRD spectra of different specimens. As can be seen from the comparative results in Figure 8, there were no other phases produced after treatment with Micro-LSP at different energies. From the XRD patterns, a lack of symmetry was observed with peaks at different energies. The asymmetry of peaks was observed since the change at the surface. Similar stories have been reported previously. Praveen et al. [34] investigated the effect of LSPwC on AM80 alloy. He pointed out that no new phases were created after reinforcement. Park et al. [35] studied the influence of the LSPwC process on the fretting corrosion of copper contacts. Moreover, he indicated that no phase transformation was detected after treatment.

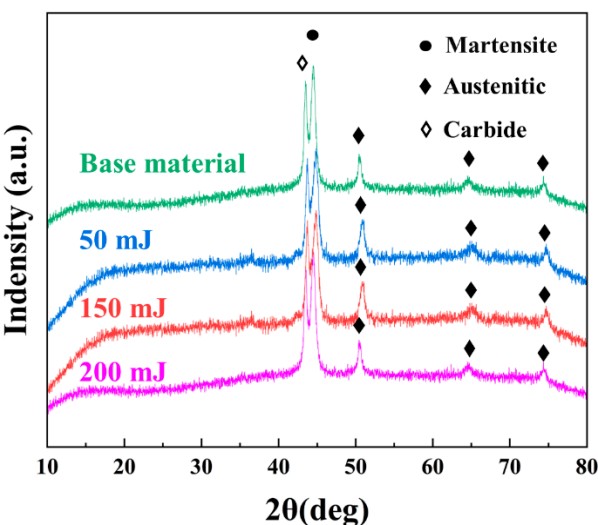

**Figure 8.** XRD patterns after different energy treatments.

### 3.5. Worn Tracks Analysis

Figure 9 shows the surfaces with different pulse energy treatments, and long strips of abrasive marks can be observed; each image also shows the specimen after wear in the lower left corner. Periodic dimples can also be observed around the abrasion marks of the strengthened specimens. Unlike the abrasion marks of the substrate, the edges of the Micro-LSP-reinforced abrasion marks were filled with a jagged, periodic pattern, which was the dimple tissue produced by Micro-LSP strengthening, but not completely worn out. Several long abrasions of varying thicknesses were also found in the center.

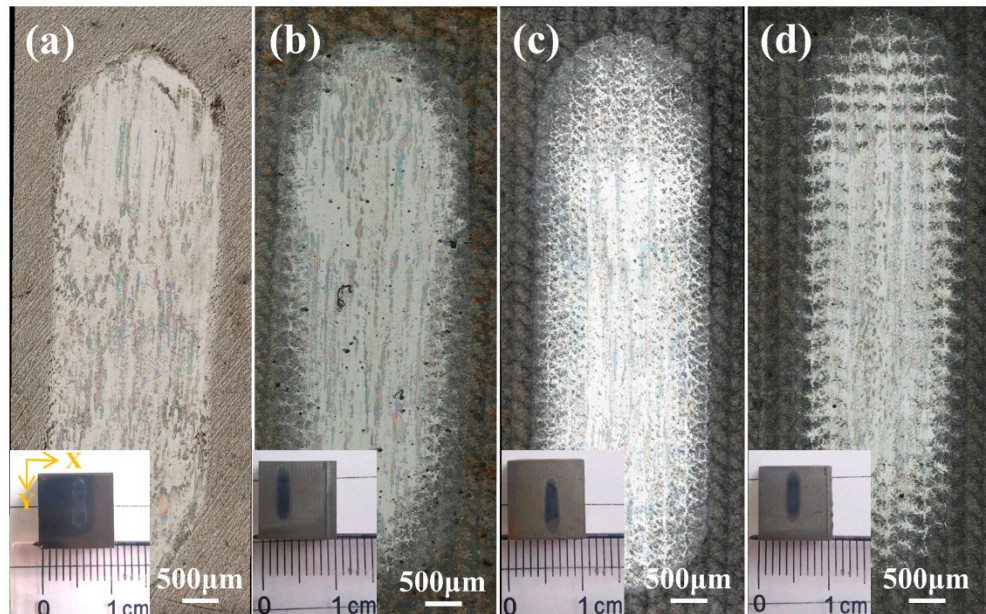

**Figure 9.** OM morphologies of wear and tear. (**a**) base material; (**b**) 50 mJ; (**c**) 150 mJ; (**d**) 200 mJ.

Figure 10a shows the surface of a specimen after abrasion with a 150 mJ pulse energy treatment, and Figure 10b shows its EDS plot. In it, the element silicon was found to be heavily distributed in the wear marks, which was due to the presence of silicon in the friction sub Si$_3$N$_4$. Observation area I in Figure 10c reveals a large amount of oxygen and silicon elements, which indicate that oxidative wear and adhesive wear occurred during the wear process. It was well understood that during the relative movement of the sub-friction and AISI 9310 steel, the two come into contact with each other, and the micro-convexities on the surface caused plastic deformation of each other. In the process of continuous movement, the surface oxide film produced by friction and high temperatures gradually breaks down, and adhesion occurs on the contact surface. In addition, the degree of oxidative wear was related to the increase in temperature. It could be explained by Archard's law [36]:

$$T = \frac{f(\pi H)^{1/2}}{8\theta_c} F_N V \tag{1}$$

where $T$ is the elevated temperature, $f$ is the coefficient of friction, $H$ is the hardness, $\theta_C$ is the thermal conductivity, $F_N$ is the load, and $v$ is the sliding speed. In the sliding experiment, $\theta_C$, $F_N$, and $v$ were all fixed values. As a result, high-hardness specimens subjected to Micro-LSP reinforcement easily reached higher temperatures, which may promote the formation of oxide films. The oxide film adhered to the surface for a long time, and acted as protection and lubrication. This inhibited adhesive wear and plastic deformation. If the sliding action continued at this point, shearing would occur at the point of adhesion, and the hard protrusions on the surface would cause the material to fall off, resulting in abrasive wear. The absence of elemental Fe in the EDS plot of Figure 10d was evidence that abrasive wear had occurred at this point. It was the same as that reported by Yu [37] et al. Figure 10e shows the SEM image of the edge of the abrasion mark, and in Figure 10f the wear details can also be observed: part of the rough multilayer porous structure was not worn completely at the edge of the abrasion mark, while the rest was worn, and the oxide layer was worn away.

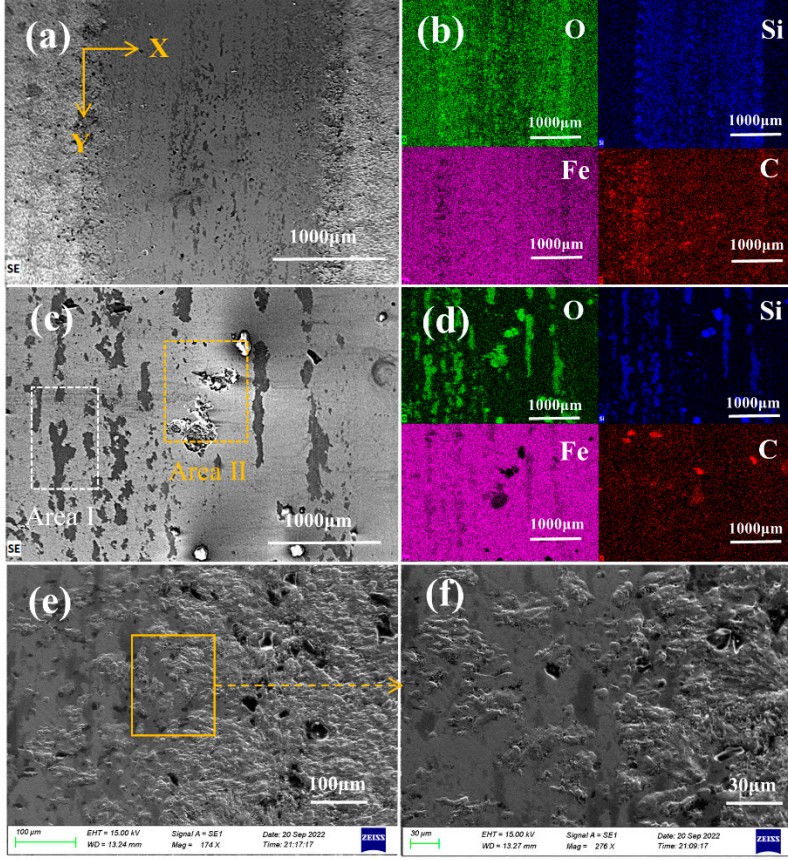

**Figure 10.** SEM and EDS morphologies after Micro-LSP treatments of 150 mJ (**a**,**b**), middle of the abrasion (**c**,**d**), and central abrasion abraded edges (**e**,**f**).

## 4. Discussion of Wear Resistance Enhancement

To quantitatively describe the enhancement of the wear resistance of AISI 9310 steel by Micro-LSP technology, Figure 11 shows the 3D morphologies, profile data, friction coefficients, and wear rates of the specimens after wear for different energy treatments. Figure 11a–d show the untreated, 50 mJ, 150 mJ, and 200 mJ treated surfaces after wear. The Micro-LSP treated specimens showed shallow depths and smaller areas of abrasion compared to the substrate. To analyze the effect of different reinforcement energies on the size of the abrasion marks, the depth profile of the section along the white dashed line is shown in Figure 11e, and the length profile along the yellow dashed line is shown in Figure 11f. The results show a reduction in the depth and length of the abrasion marks on the Micro-LSP treated specimens compared to the untreated specimens. The smaller ones were particularly noticeable after the 200 mJ treatment, where the wear depth was reduced from 5.2 μm to 3.8 μm (a reduction of approximately 26.9%), and the wear length from 7112 μm to 6587 μm. In addition, as shown in Figure 11e, all the edges of the abrasion marks form bumps. These are due to the plastic flow that occurs during sliding wear when the material was squeezed and sheared against the friction substrate. The untreated substrate material had a greater bump height than the Micro-LSP-treated specimens. Additionally, the height of the bulge decreased with increasing pulse energy. These phenomena stem from the fact that lasers with high pulse energies lead to more pronounced process hardening effects. Although the higher energy of the laser increases the hardness more, it also makes the material less plastic. Lower plasticity results in poorer flow of the metal, i.e., the work-hardened surface inhibits the ability of the material to recover from deformation. Figure 11g shows the changes in the COFs after treatment with different pulse energies. During the initial phase, the curve oscillated and fluctuated in a wide range, eventually leveling off. The reason for the fluctuations can be understood as a

lack of stability on the surface of the Micro-LSP-treated samples. These structures increased the surface roughness, and also made the friction coefficient unstable. As the wear test progressed, the rough dimpled structure was worn away; then, it went to the hardened layer. The high hardness of the hardened layer had a more stable structure and properties compared to the dimpled structure. This resulted in a gradual stabilization of the COF. In addition, the time required to enter the steady state varied with the COF of the specimen after treatment with different pulse energies. The higher the processing energy, the more time it took to enter the steady state. This may have been due to the different heights of dimpled structures produced by different pulse energies (Figure 4i). There was also a small oscillation after the COF curve smoothed out. The reason for this may be due to a large number of residues, such as abrasive chips, that were generated during contact with the friction substrate during the wear test. Such residues caused sudden shocks in the COF. In Figure 11h, the wear rates of all experiments were calculated. The wear rates of the 50 mJ, 150 mJ, and 200 mJ treated specimens were reduced by 56.1%, 65.1%, and 74.2%, compared to untreated specimens, respectively. A low wear rate means high wear resistance of the material [38,39]. As mentioned above, Micro-LSP significantly reduced the COF and wear surface profile of AISI 9310 steel, demonstrating its effectiveness in improving friction and wear properties.

To qualitatively understand the mechanism of improved wear resistance with Micro-LSP technology, Figure 12 shows a comparison of the wear process of the specimen before and after strengthening was carried out. The surface microstructure and gradient hardening layer are essential for improvement to wear resistance during the friction process. The EDS results (Figure 6) show that the Micro-LSP technique induces an oxide layer on the surface of the specimen. According to previous reports, the initial oxide layer acted as a lubricant to reduce the coefficient of friction [40]. In addition, the dimple-structured oxide layer that formed on the surface had a high degree of hardness. This made it even better at resisting foreign intrusion or wear and tear [41]. The sliding friction of the friction pair was effectively limited by the oxide layer, resulting in a lower wear rate. The use of laser-induced oxide layers has also been suggested to reduce wear in the report by Harlinet et al. [38]. Moreover, SEM (Figure 5e–h) and OM (Figure 4) results that reinforcement induced dimpled structures and hardened layers on the surface of the sample, resulting in an increase in roughness and hardness at the initial stage of wear, compared to untreated samples. The contact state of the sample was altered due to the structure of dimples induced on the surface by Micro-LSP (Figure 12b). The smaller contact area results in a lower coefficient of friction than the substrate. As the tips of the dimples were gradually ground away, the abrasive chips gradually fell back into the recesses between the structures. The structure, in turn, acted as a collection of abrasive chips. The grinding chips were located in the structural recesses and could be rolled relative to the friction substrate. The coefficient of friction was reduced by converting part of the sliding friction into rolling friction. The deeper the dimpled structure was, the greater the number of abrasive chips that could be stored (Figure 4i). Thus, the coefficient of friction was reduced as much as possible. In addition, the instability of the contact between the abrasive chips and the tip of the structure caused the COF to oscillate at the earliest onset of wear. When the dimpled structure became completely ground out, the high hardness gradient hardening layer continuously resisted the intrusion of frictional substrates, and kept the COF stable. On the other hand, there was no black-bound layer protection during the Micro-LSP treatment, thus the thermal effects generated during the strengthening process could not be absorbed in time. This resulted in the surface of the specimen being ablated and losing its metallic luster after Micro-LSP strengthening (Figure 2b). This was determined by the pulse duration of the Micro-LSP technology. Unlike the pulse width of the Femtosecond-Laser Shock Peening technology ($10^{-15}$ s), the pulse duration of the Micro-LSP technology was approximately $10^{-9}$ s. The duration of the pulse greatly influences the thickness of the heat-affected layer. The time scale for the thermal processes arising from electron–phonon interactions is about $10^{-11}$ s to $10^{-12}$ s. When the duration of the laser pulse is greater than the thermal process

time, a thermal effect is produced; otherwise, it does not. The pulse time of the Micro-LSP is less than the thermal process time, so there is enough time for the electrons to absorb heat during intensification. Subsequently, the electrons transfer energy into the lattice by coupling with phonons, or in free collisions. A sufficiently long pulse time will allow the heat generated at the surface to be transferred to the interior of the specimen [42]. In addition, large pulses of energy cause the electrons to carry more energy. This increases the thermal effect on the surface, which in turn produces a sufficiently thick melting layer. Thicker remelting layers result in increased surface roughness. In this way, the COF was reduced and the wear resistance was improved.

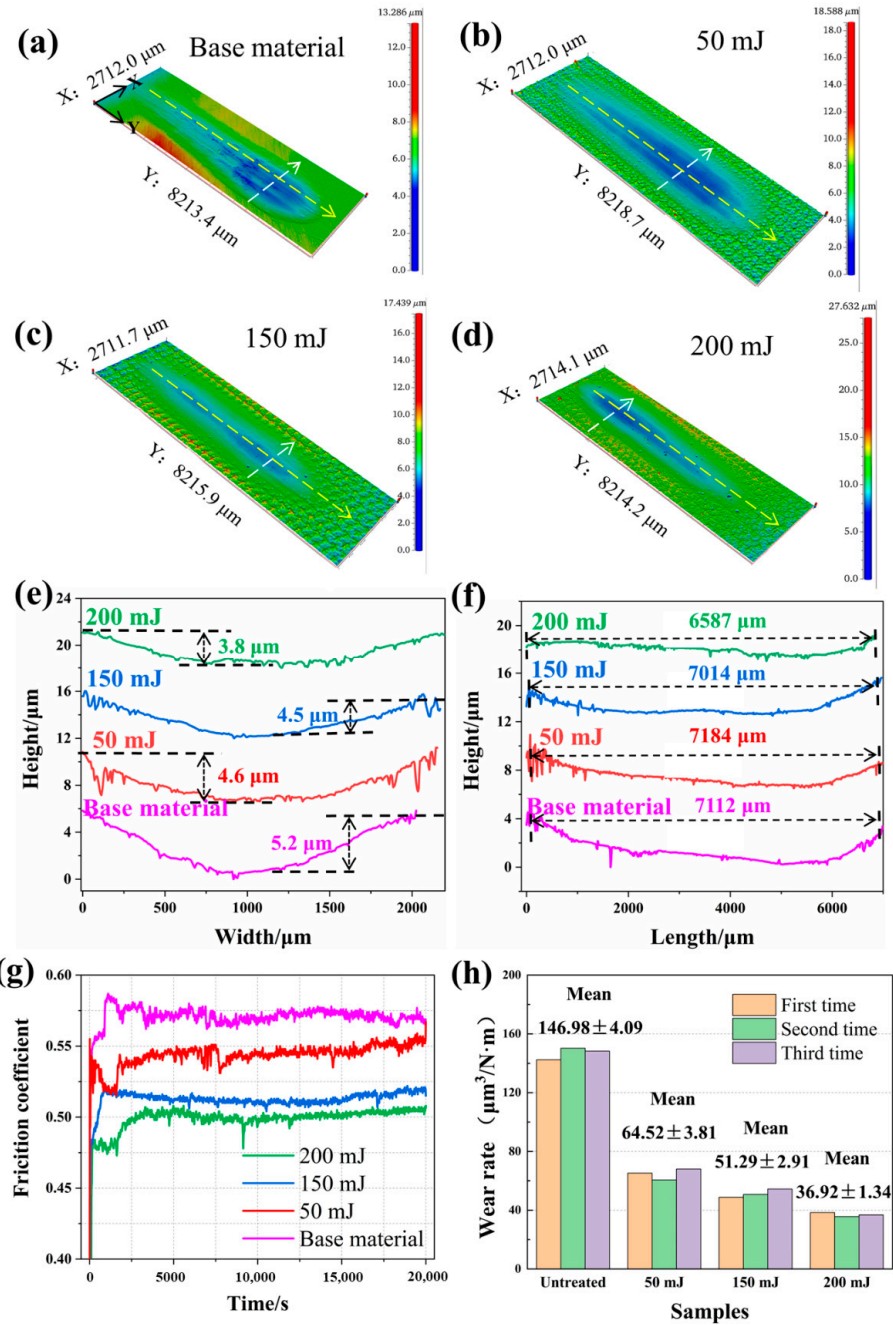

**Figure 11.** Abraded specimens with different energy treatments. (**a**–**d**) 3D morphologies; (**e**,**f**) abrasion contours; (**g**) friction coefficients; (**h**) wear rates.

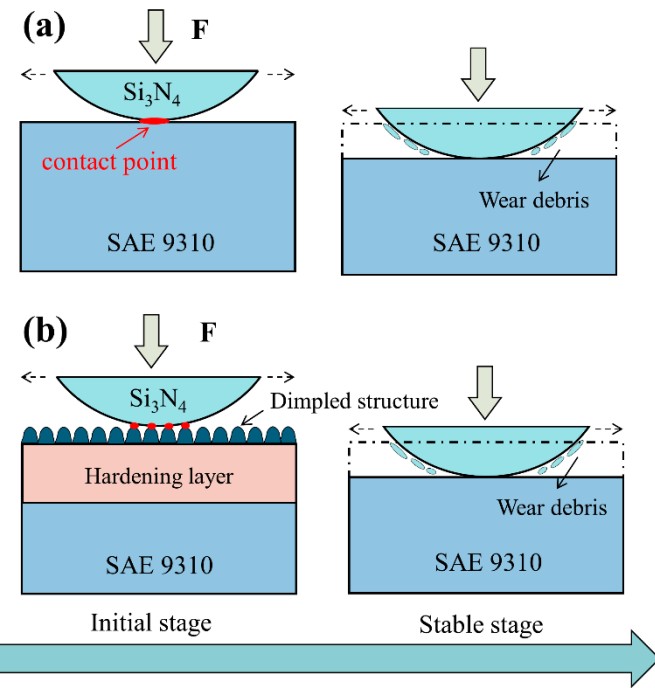

**Figure 12.** Schematic illustrations of the wear resistance improvement mechanism of Micro-LSP (**a**) pre-strengthening; (**b**) after strengthening.

The EDS surface sweep results showed that there was a significant amount of oxygen in the remelted layer; this was because the oxygen element in the water curtain was also participating in the reaction as the resulting thermal effect melted the surface, curing on the surface of the specimen after cooling. In addition to this, Micro-LSP technology offers the advantages of high machining accuracy and efficiency; the increase in hardness is also greater. Song et al. [43] found an increase in hardness of approximately 25%, from 236.7 $HV_{0.05}$ to 297.4 $HV_{0.05}$, when they used the LSP technique to reinforce AISI 9310 steel. In the present study, however, the increase in hardness was up to approximately 50% (Figure 7). Micro-LSP technology offers completely new approaches and possibilities for hardness improvement. The Micro-LSP-treated specimens had a lower COF compared to the base material, and this suggests some new wear mechanisms. In frictional wear processes, Yu et al. [17] argued that the value of COF was closely related to the actual surface contact/state contact interface, as shown in Figure 4i with the OM results, and with the SEM results in Figure 5. The Micro-LSP treatment induced dimples on the surface, and neatly arranged them on the surface of AISI 9310 steel. It was different from the surface morphology produced by LSP reinforcement, and had not been reported in this context. These laser-induced dimpled structures can form a sliding interlayer between the friction substrate and the substrate. The structure can significantly reduce the contact area when the friction process occurs. This results in a reduced COF and, thus, improved wear resistance.

## 5. Conclusions

In this study, AISI 9310 steel was treated with Micro-LSP at different pulse energies. The effects of the laser energy on the surface morphology, mechanical properties, and wear behavior were investigated. The following conclusions can be drawn from the analysis and observations:

(1)  Micro-LSP treatment significantly increased the surface roughness of the specimen due to laser ablation. Periodically arranged dimpled structures were formed on the surface, and the depth of the structure was strongly influenced by the different pulse energies. Large energies produced thermal effects that caused the surface to lose its metallic luster, and produced a thick remelted oxide layer.

(2)     After the Micro-LSP treatment, the surface microhardness increased, from a maximum of 827 $HV_{0.5}$ to 1228 $HV_{0.5}$, an increase of approximately 50%. The hardness was distributed in a gradient along the depth direction. The maximum impact range was approximately 600 μm.

(3)     After strengthening, the COF was reduced from 0.56 to approximately 0.5 at maximum, a reduction of 12%. The abrasion depth was reduced from a maximum of 5.2 μm to 3.8 μm, a reduction of approximately 25%.

(4)     The Micro-LSP treatment increased the wear resistance of the material by approximately 50% to 70%. This was probably due to the combined effect of the introduction of a dimpled structure onto the surface, and the hardening layer. The dimpled structure reduced the COF by reducing the contact area and storing abrasive chips. The gradient hardening layer continuously resisted wear and tear to keep the COF stable.

**Author Contributions:** Conceptualization, L.Z. and X.L.; methodology, L.Z.; validation, X.L. and T.Z.; formal analysis, L.Z. and X.L.; investigation, L.Z. and X.L.; resources, L.Z., P.L. and X.P.; data curation, X.L.; writing—original draft preparation, L.Z. and X.L.; writing—review and editing, P.L. and X.P.; visualization, X.L. and X.P.; supervision, L.Z.; project administration, L.Z. All authors have read and agreed to the published version of the manuscript.

**Funding:** This research was funded by "The National Major Science and Technology Project of China, grant number J2019-IV-0014-0082" and "The National Natural Science Foundation of China, grant number 51875574".

**Institutional Review Board Statement:** Not applicable.

**Informed Consent Statement:** Not applicable.

**Data Availability Statement:** The data used to support the findings of this study are available from the corresponding author upon request.

**Acknowledgments:** The authors thank the anonymous reviewers for their critical and constructive review of the manuscript. This research was supported by the National Major Science and Technology Project of China [grant number J2019-III-0008] [grant number J2019-IV-0014-0082]; and the National Natural Science Foundation of China [grant number 51875574].

**Conflicts of Interest:** The authors declare no conflict of interest.

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
