# Peer review of "Research on Wear Resistance of AISI 9310 Steel with Micro-Laser Shock Peening"

_metals, doi:10.3390/met12122157_

Round 1

Reviewer 1 Report

1.     The authors mention that Micro-LSP technology is newly proposed. However, the definition of the Micro-LSP is not clear. Please mention the definition of Micro-LSP. This kind of research has been done in the past. Please cite them appropriately. For example, there are the following papers:

doi.org/10.3390/met10010152

doi.org/10.1016/j.optlastec.2016.02.007

2.     There is no discussion of the effect of residual stresses introduced by LP on wear characteristics. Please discuss this point as well.

3.     Please describe whether the coating was applied during laser peening.

4.     There is only one wear test result for each test condition. For more reasonable conclusions, I believe that experiments with different loads and durations are needed.

5.     Please add the dimensions of the counterpart material, silicon nitride, and the lubrication conditions in the wear test conditions. I think that a description of the Hertzian load is necessary.

6.     The description in the manuscript is very confusing. There are many places where the subject and predicate of the sentence do not correspond. It is recommended that the authors review the English.

7.     In some of the figures, the legends are too small to be legible. In particular, the following figures should be enlarged: Fig.2c, 4e-i, 11.

8.     Please show the relationship between the direction of laser peening (Fig.2c) and the sliding direction (Fig.3, 4, 5, 6, 9, 10, 11). Include the coordinate axes in each figure.

9.     It is necessary to correct the mis-numbering in Figures 9 and 10.

10.   The color of the graph should be consistent with the specimen in Fig. 11 e, f and g.

11.   Please make the scale in the figure easier to read in Fig.4,5,6,10.

12.   The authors stated that “The results showed that the Micro-LSP treatment improved the surface roughness of AISI 9310 steel.” in line 142-143. This statement is misleading.

Reviewer 2 Report

This paper  examined the new LSP technique: Micro-LSP technology for surface modification and strengthening of AISI 9310 steel. These results and discussion have highly values for the advances of LSP.

(1)"a new LSP processing method: Micro-Laser Shock Peening (Micro-LSP) technology,which allows for more efficient processing and better accuracy than LSP technology. "

Please explain in detail the difference between the new LSP processing method and conventional methods.

(2)

"2.1. Experimental material and processing parameters
The sample size was 10 mm×10 mm×5 mm and was obtained from a commercial AISI 9310 steel board by wire-cutting method (Ming Hang, China). Both sides of AISI 9310 steel
had been vacuum carburized twice, each time for 6 h. "

The author should write in detail about the vacuum carburizing method and the thickness of the carburized layer.

(3)How about discussing the results of XRD measurements of the crystal structure of the surface after the laser treatments? It would greatly affect the wear resistance , such as whether martensite remains.

Reviewer 3 Report

The numbering of the figures is completely wrong. Oxygen analysis using the EDS method is questionable, especially for surfaces, and the given accuracy of the obtained result is certainly within the measurement error range. Incorrect description of the legend in the graph in figure 7b base material and 200 mJ are swapped. 

Round 2

Reviewer 1 Report

Thank you for revising the manuscript.

The authors have made appropriate corrections in a short period of time.